# Impact of Visual Design Elements and Principles in Human Electroencephalogram Brain Activity Assessed with Spectral Methods and Convolutional Neural Networks

**DOI:** 10.3390/s21144695

**Published:** 2021-07-09

**Authors:** Francisco E. Cabrera, Pablo Sánchez-Núñez, Gustavo Vaccaro, José Ignacio Peláez, Javier Escudero

**Affiliations:** 1Department of Languages and Computer Sciences, School of Computer Science and Engineering, Universidad de Málaga, 29071 Málaga, Spain; fecabrera@uma.es (F.E.C.); fabianvaccaro@uma.es (G.V.); jipelaez@uma.es (J.I.P.); 2Centre for Applied Social Research (CISA), Ada Byron Research Building, Universidad de Málaga, 29071 Málaga, Spain; 3Instituto de Investigación Biomédica de Málaga (IBIMA), 29071 Málaga, Spain; 4Department of Audiovisual Communication and Advertising, Faculty of Communication Sciences, Universidad de Málaga, 29071 Málaga, Spain; 5School of Engineering, Institute for Digital Communications (IDCOM), The University of Edinburgh, 8 Thomas Bayes Rd, Edinburgh EH9 3FG, UK

**Keywords:** EEG, emotion classification, CNN, spectral analysis, visual perception, visual attention, visual features, visual design elements and principles (VDEPs)

## Abstract

The visual design elements and principles (VDEPs) can trigger behavioural changes and emotions in the viewer, but their effects on brain activity are not clearly understood. In this paper, we explore the relationships between brain activity and colour (cold/warm), light (dark/bright), movement (fast/slow), and balance (symmetrical/asymmetrical) VDEPs. We used the public DEAP dataset with the electroencephalogram signals of 32 participants recorded while watching music videos. The characteristic VDEPs for each second of the videos were manually tagged for by a team of two visual communication experts. Results show that variations in the light/value, rhythm/movement, and balance in the music video sequences produce a statistically significant effect over the mean absolute power of the Delta, Theta, Alpha, Beta, and Gamma EEG bands (*p* < 0.05). Furthermore, we trained a Convolutional Neural Network that successfully predicts the VDEP of a video fragment solely by the EEG signal of the viewer with an accuracy ranging from 0.7447 for Colour VDEP to 0.9685 for Movement VDEP. Our work shows evidence that VDEPs affect brain activity in a variety of distinguishable ways and that a deep learning classifier can infer visual VDEP properties of the videos from EEG activity.

## 1. Introduction

### 1.1. Overview

Visual communication design is the process of creating visually engaging content for the creative industries, to communicate specific messages and persuade in a way that eases the comprehension of the message and ensures a lasting impression on its recipient. Therefore, the main objective of visual communication design as a discipline, is the creation of visually attractive content that impacts the target audience through persuasion, the stimulation of emotions and the generation of sentiments. This objective is achieved through conscious manipulation of the inherent elements and principles, such as shape, texture, direction harmony, and colour that make up the designed product [1].

The elements of visual design describe the building blocks of a product’s aesthetics, and the principles of design tell how these elements should go together for the best results. Although there is no consensus on how many visual design elements and principles (VDEPs) there are, some of these design reference values have been thoroughly studied for centuries and are considered the basis of modern graphic design [2]. One of the most important breakthroughs in graphic design history is the theory of the Gestalt in the 1930s, which studied these VDEPs from a psychological point of view, and discovered that the notion of visual experience is inherently structured by the nature of the stimulus triggering it as it interacts with the visual nervous system [3]. Since then, several studies have explored the psychological principles of different VDEPs and their influence on their observers [4,5,6]. As an example, in colour theory, some colours have been experimentally proven to convey some semantic meanings, and they are often used to elicit particular emotions in the viewers of visual content [7,8,9].

Within this context, in recent years diverse research has been carried out to deepen our understanding of brain activity and visual perception [10]. Research findings on visual perception have been applied to real-world applications in a wide range of fields, including visual arts, branding, product management, and packaging design to increase consumer spending, creating engaging pedagogic content, and survey design, among others [11,12,13,14]. These works mostly focus on the VDEPs capability of eliciting specific emotions in the viewer and how to trigger the desired response from the viewer, such as favouring the learning of content or increasing the probability of purchasing some goods. Multiple approaches have recently addressed the study of human visual perception from within the fields of attention and perception psychology and computational intelligence with studies focused on the neuropsychology of attention and brain imaging techniques and their applications (fMRI or electroencephalogram, EEG) [14,15], pupillometry [16] or eye-tracking [17,18] as a scanning technique of information processing as well as neuroaesthetics [19], computational neuroscience [20], computational aesthetics [21], or neuromarketing, and consumer neuroscience studies [22,23], among others [24].

However, it is not yet known how the VDEPs impact human brain activity. In this regard, we turn our attention to the DEAP dataset [25], a multimodal dataset for the analysis of human affective states with EEG recordings and synchronous excerpts of music videos. Previous research carried out with DEAP dataset pursued emotion recognition from the EEG records by using 3D convolutional neural networks [26], a fusing of learned multi-modal representations and dense trajectories for emotional analysis in videos [27], studying arousal and valence classification model based on long short-term memory [28], DEAP data for mental healthcare management [28], or accurate EEG-based emotion recognition on combined features using deep convolutional neural networks, among others [29].

Beyond the information about levels of valence, arousal, like/dislike, familiarity, and dominance in the DEAP dataset, we sought to analyse the EEGs and videos to extract some of the most important VDEPs—Balance, Colour, Light, and Movement—and search for measurable effects of these on the observable EEG brain activity. The Balance, Colour, Light and Movement principles are fundamental for professional design pipelines and are universally considered to influence the emotions of the viewer, as designers rely on the contrasts, schemes, harmony, and interaction of colours to evoke reactions and moods in those who look at their creations [30,31,32,33].

The Balance refers to the visual positioning or distribution of objects in a composition, gives the visual representation of a sense of equality, and can be achieved through a symmetrical or asymmetrical composition to create a relationship of force between the various elements represented, either to compensate or, to the contrary, to decompensate the composition [6,34]. We speak of symmetrical balance when the elements are placed evenly on both sides of the axes. Secondly, the Colour helps communicate the message by attracting attention, setting the tone of the message, guiding the eye to where it needs to go and determines 90% of the choice of a product [35,36]. The warmth or coldness of a colour attends to subjective thermal sensations; it can be cold or warm depending on how it is perceived by the human eye and the interpretation of the thermal sensation it causes [36]. Thirdly, the Light or “value” refers to relative lightness and darkness and is perceived in terms of varying levels of contrast; it determines how light or dark an area looks [34,37]. Finally, Movement refers to the suggestion of motion using various elements and it is the strongest visual inducement to attention. Different studies have dissected the exact nature of our eye movement habits and the patterns our eyes trace over when viewing specific things [17,18]. Strongly connected with direction VDEPs, rhythm/movement provide designers with the chance to create final pieces with good flow from top to bottom, left to right, corner A to corner B, etc., [37]. By layering simple shapes of varying opacities, an abrupt change in the camera, a counterstroke or a tracking shot, or a sudden change in the subject’s action, it is possible to create a strong sense of speed and motion and determine an effect of mobility or immobility.

### 1.2. Background

#### 1.2.1. EEG Symmetry Perception

Symmetry is known to be an important determinant of aesthetic preference. Sensitivity to symmetry has been studied extensively in adults, children, and infants with diverse research ranging from behavioural psychology to neuroscience. Adults detect symmetrical visual presentations more quickly and accurately than asymmetrical ones and remember them better [38,39]. The perception and appreciation of visual symmetry have been studied in several EEG/fMRI experiments, some of the most recent studies are focused on symmetric patterns with different luminance polarity (anti-symmetry) [40], the role of colour and attention-to-colour in mirror-symmetry perception [41], or contour polarity [42], among others.

#### 1.2.2. EEG Colour Perception

The perception of colour is an important cognitive feature of the human brain and is a powerful descriptor that considerably expands our ability to characterize and distinguish objects by facilitating interactions with a dynamic environment [43]. Some of the most recent studies that have addressed the interactions of colour and human perception through EEG are the response of a human visual system to continuous colour variation [44], human brain perception and reasoning of image complexity for synthetic colour fractal and natural texture images via EEG [45], or neuromarketing studies based on the study of colour perception and the application of EEG power for the prediction and interpretation of consumer decision-making [46], and so forth.

#### 1.2.3. EEG Brightness Perception

Brightness is one of the most important sources of perceptual processing and may have a strong effect on brain activity during visual processing [47]. Brain responses of the brain to changes in brightness were explored in different studies centred on the luminance and spatial attention effects on early visual processing [48], the grounding valence in brightness through shared relational structures [49] or the interaction of brightness and semantic content in the extrastriate visual cortex [50].

#### 1.2.4. EEG Visual Motion Perception

Detecting the displacement of retinal image features has been studied for many years in both psychophysical and neurobiological experiments [51]. Visual motion perception has been explored through EEG technique in several lines of research such as the speed of neural visual motion perception and processing in the visuomotor reaction time of young elite table tennis athletes [52], visual motion perception for prospective control [53], or visual perception of motion to cortical activity, by evaluation of the association of quantified EEG parameters with a video film projection [54], the analysis of neural responses to motion-in-depth using EEG [55], or the examination of the time course of motion-in-depth perception [56].

In addition, recent research has sought to delve deeper into Brain-Computer Interfaces (BCI), an emerging area of research that aims to improve the quality of human-computer applications [57], and the relationship with Steady-State Visually Evoked Potentials (SSVEPS), a stimulus-locked oscillatory response to periodic visual stimulation commonly recorded in an electroencephalogram (EEG) studies in humans, through the use and execution of Convolutional Neural Networks (CNN) [58,59]. Some of these research works have focused on the review of the steady-state evoked activity, its properties, and the mechanisms of SSVEP generation, as well as the SSVEP-BCI paradigm and the recently developed SSVEP-based BCI systems [60]; studies focused on analysing Deep Learning-based classification for BCI through comparisons among various traditional classification algorithms to the newer methods of deep learning, exploring two different types of deep learning methods: CNN and Recurrent Neural Networks (RNN) with long short-term memory (LSTM) architecture [61]. In addition, recent research works proposed a novel CNN for the robust classification of an SSVEPS paradigm, where the authors measured electroencephalogram (EEG)-based SSVEPs for a brain-controlled exoskeleton under ambulatory conditions in which numerous artefacts may deteriorate decoding [62]. Another work that analyses SSVEPS by using CNNs is the work of S.Stober et al. [63], which analysed and classified EEG data recorded within a rhythm perception study. In this last case, the authors investigated the impact of the data representation and the pre-processing steps for this classification task and compared the different network structures.

### 1.3. Objective

It is not known whether predictors can be constructed to classify these VDEPs based on brain activity alone. The large body of previous research explored the effects of the VDEPs as isolated features, even though during human visual processing and perception many of them act simultaneously and are not appreciated individually by the viewer in different situations.

In this work, we present a novel methodology for exploring this relationship between VDEPs and brain activity in the form of EEGs available on the DEAP dataset. Our methodology consisted of combining the expert recognition and classification of VDEPs, statistical analysis, and deep learning techniques, which we used to successfully predict VDEPs solely from the EEG of the viewer. We tested whether:➢There is a statistical relationship between the VDEPs of video fragments and the mean EEG frequency bands (δ, θ, α, β, γ) of the viewers.➢A simple Convolutional Neural Network model can accurately predict the VDEPs in video content from the EEG activity of the viewer.

## 2. Materials and Methods

### 2.1. DEAP Dataset

The DEAP dataset is composed of the EEG records and peripheral physiological signals of 32 participants, which were recorded as each watched 40 1-min-long excerpts of music videos, relating to the levels of valence, arousal, like/dislike, familiarity, and dominance reported by each participant. Firstly, the ratings came from an online self-assessment where 120 1-min extracts of music videos were rated by volunteers based on emotion classification variables (namely arousal, valence, and dominance) [25]. Secondly, the ratings came from the participants’ ratings on these emotion variables, face video, and physiological signals (including EEG) of an experiment where 32 volunteers watched a subset of 40 of the abovementioned videos. The official dataset also includes the YouTube links of the videos used and the pre-processed physiological data (down sampling, EOG removal, filtering, segmenting, etc.) in MATLAB and NumPy (Python) format.

DEAP dataset pre-processing:The data were down sampled to 128 Hz.EOG artefacts were removed.A bandpass frequency filter from 4 to 45 Hz was applied.The data was averaged to the common reference.The EEG channels were reordered so that all the samples followed the same order.The data was segmented into 60-s trials and a 3-s pre-trial baseline was removed.The trials were reordered from presentation order to video (Experiment_id) order.

In our experiments, we use the provided pre-processed EEG data in NumPy format, as recommended by the dataset summary, since it is especially useful for testing classification and regression techniques without the need of processing the electrical signal first. This pre-processed signal contains 32 arrays corresponding to each of the 32 participants, with a shape of 40 × 40 × 8064. Each array contains data for each of the 40 videos/trials of the participant, signal data from 40 different channels (the first 32 of them being EEG signals and the remainder 8 being peripheral signals such as temperature and respiration) and 8064 EEG samples (63 s × 128 Hz). As we are not working with emotion in this work, the labels provided for the videos on the DEAP dataset were not used in this experiment. Instead, we retrieved the original videos from the URLs provided and performed an exhaustive classification of them on the studied VDEPs. Of the original 40 videos, 14 URLs pointed to videos that were taken down from YouTube as of the 4 June 2019, therefore, only 26 of the videos used in the original DEAP experiment were retrieved and classified. The generated dataset was updated on the 5 May 2021.

### 2.2. VDEP Tagging and Timestamps Pre-Processing

The researchers retrieved the 26 1-min videos from the original DEAP experiment. These video clips accounted for 1560 1-s timestamps. These videos were presented to two experts on visual design, who were tasked to tag each second of video on the studied VDEPs (Figure A1, Figure A2, Figure A3 and Figure A4), considering the following labels:○Colour: “cold” (class 1), “warm” (class 2) and “unclear”.○Balance: “asymmetrical” (class 1), “symmetrical” (class 2) and “unclear”.○Movement: “fast” (class 1), “slow” (class 2) and “unclear”.○Light: “bright” (class 1), “dark” (class 2) and “unclear”.

The timestamps tagged as “unclear” or where experts disagreed were discarded. On the other hand, the number of timestamps per class was unbalanced, because the music videos exhibited different visual aesthetics and characteristics among different sections within the same videos; for example, most of the timestamps were tagged as asymmetrical. Therefore, we computed a sub-sample for each VDEP, selecting *n_v_* random timestamps, where *n_v_* represents the number of cases in the smaller class. The resultant number of timestamps belonging to each class is displayed in Table 1. It is important to notice that all the participants provided the same amount of data points, therefore each second from Table 1 corresponds to 32 epochs, one for each participant. The four VDEPs are being considered as independent of each other, therefore, each second of video will appear classified only once in each of the four VDEPs, either within one of the two classes or as an unclear timestamp. The VDEPs tagging of the selected video samples from the DEAP dataset is available in Appendix A (Table A11).

Since the DEAP dataset provides the EEG signal and the experts provided the VDEP timestamps for the videos used in the original experiment, an extraction-transformation process (ETP) was performed. In this process, we obtained a large set of 1-s samples from the DEAP dataset classified according to their VDEPs. The steps of this ETP are shown in Figure 1.

This work uses statistical analysis as well as convolutional neural networks to analyse the relationship between VDEPs and the EEG, so we designed the ETP to return the outputs in two different formats, readily available to perform each of these techniques. The first output is a set of spreadsheets containing the EEG bands for the studied samples and their VDEP classes, which are used to find statistical relationships. The second output is a set of NumPy matrices with the shape of 128 × 32, each containing the 32 channels of a second of an EEG signal at 128 Hz and classified by their VDEP class that we used to train and test the artificial neural networks. This ETP was repeated four times, one for each VDEP studied in this work.

The first step of the ETP is the extraction of the intervals from the DEAP dataset according to the VDEP timestamps provided by the experts. In this step, we must take into consideration that the first 3 s of the EEG signal for each video trial in the DEAP dataset corresponds to a pre-trial for calibration purposes and therefore must be ignored when processing the signals. We also discard the channels numbered 33–40 in the DEAP dataset since they do not provide EEG data but peripheral signals, which are out of the scope of this work.

In the second step, we trimmed the first and last 0.5 s of each interval, and in the third step, we split the remainder of the signal in 1-s intervals. Given that the visual design experts extracted the VDEPs from the videos by 1-s segments, the removal of a full second of each interval (half a second on each end) strengthens the quality of the remaining samples by reducing the truncation error. An epoch is one second of a subject’s EEG with information about the VDEPs that is currently being displayed. When extracting epochs, subjects are not considered separately. For the authors, the subject has attributed a series of samples, as well as the rest of the subjects, to finally put them all together and obtain a set of epochs independently of the subject they come from. Therefore, from each second of the video, we extract 32 epochs (one for each subject). The samples are extracted from the remainder of the interval by taking 1-s samples, this approach mitigates the effects of the perception time, which is the delay between the occurrence of the stimuli and its perception by the brain [64]. In Figure 2 we show the resulting samples obtained from a 5-s interval classified in the same class.

These samples can be represented as (128 × 32) matrices and can be used to obtain EEG characteristics such as the delta, theta, alpha, beta, and gamma channels that we used in our statistical analysis. In our case, we used a basic FFT (Fast-Fourier Transform) filtering function to extract these time-domain waveforms [65].

The second output was used to train artificial neural networks (ANN), so in the ETP we normalized the values of the samples from their original values to the [0, 1] interval by using Equation (1), as normalized inputs have proven an increase on the learning rate of convolutional neural networks [66].
(1)$$f\left(x\right)=\frac{x-\min\left(X\right)}{\max\left(X\right)-\min\left(X\right)}.$$

### 2.3. Convolutional Neural Network (CNN)

In this work, we used a generic CNN architecture typically used in image analysis consisting of a sequence of two-dimensional convolutional layers followed by a fully connected layer with sigmoid activation. The same network architecture was used in each of the VDEPs, and the neural network was trained using the previously extracted epochs, given that the epochs do not consider the subject from which they were extracted, it could be considered that all the subjects were equally fed into the model. The training process of the model was performed by randomly selecting 90% for the training sample and 10% for testing in a non-exhaustive 10-fold cross-validation execution. The performance metrics for the area under the ROC curve (AUC), accuracy and area under the precision-recall curve (PR-AUC) were computed for the validation set after each execution.

The input of the model re-arranged the EEG channels by proximity for improving the information of adjacent channels, which helps convolutional neural networks to learn more effectively [67]. The original ordering of the channels and the final order are shown in Figure 3, including a plot of input before and after this re-ordering.

## 3. Results

We tested the normality of the distribution of measurements of the mean power measurements across all channels (δ, θ, α, β, and γ) using the Kolmogorov-Smirnov test with the Lilliefors Significance Correction. Then, we conducted Mann-Whitney U Tests to examine the differences on each of the mean power measurements according to the categories related to each of the VDEP criteria, e.g., the differences in the Alpha band between the Symmetrical and Asymmetrical categories for the Balance VDEP. The Mann-Whitney U test (also known as the Wilcoxon rank-sum test) was chosen to test the null hypothesis that there is a probability of 0.5 that a randomly drawn observation for one group is larger than a randomly drawn observation from the other.

The Kolmogorov-Smirnov test proved that the distribution of measurements of the mean δ, θ, α, β, and γ bands of the EEGs provided in the DEAP were not normally distributed, with *p* < 0.0001 in all cases, as shown in Table A1.

### 3.1. Balance

The mean power measurements across all channels were statistically significantly higher in the Symmetrical category than the Asymmetrical category for the Balance, with *p* < 0.05 in all cases (Figure 4). The details of the ranks and statistical average of the mean band measurements for the Balance are listed in Table A2. The full Test Statistics for Mann-Whitney U Tests conducted on the Balance are listed in Table A7.

### 3.2. Colour

There were no statistically significant differences in the mean power measurements across all channels (δ, θ, α, β, and γ) between the Warm and Cold categories for the Colour, with *p* > 0.05 in all cases (Figure 5). The details of the ranks and statistical average of the mean power measurements for the Colour are listed in Table A3. The full Test Statistics for Mann-Whitney U Tests conducted on the Colour are listed in Table A8.

### 3.3. Light

All the mean power measurements across all channels (δ, θ, α, β, and γ) were statistically significantly higher in the Bright category than the Dark category for the Light, with *p* < 0.05 in all cases (Figure 6). The details of the ranks and statistical average of the mean power measurements for the Light are listed in Table A4. The full Test Statistics for Mann-Whitney U Tests conducted on the Light are listed in Table A9.

### 3.4. Movement

All the mean power measurements across all channels (δ, θ, α, β, and γ) were statistically significantly higher in the Fast category than the Slow category for the Movement, with *p* < 0.05 in all cases (Figure 7). The details of the ranks and statistical average of the mean power measurements for the Movement are listed in Table A5. The full Test Statistics for Mann-Whitney U Tests conducted on the Movement are listed in Table A10.

### 3.5. Convolutional Neural Networks (CNN)

All the VDEP targets were accurately predicted from the EEG signal using a simple CNN classification model trained using a non-exhaustive 10-fold cross-validation approach. The performance metrics were obtained from each training and the averages were computed. It is important to notice that we trained 40 independent models, 10 for each VDEP, which shared the same base structure (see Table A6). The prediction of Movement was the most accurate among the studied VDEPs (AUC = 0.9698, Accuracy = 0.9675, PR-AUC = 0.9569). On the other hand, the corresponding trained model struggled to predict the Colour VDEP from the EEG signal input (AUC = 0.7584, Accuracy = 0.7447, PR-AUC = 0.6940). A summary of the classification performance for each VDEP is displayed in Table 2.

## 4. Discussion

The VDEPs are responsible for capturing our attention, persuading us, informing us, and engaging a link with the visual information represented. The answer to what makes a good design and how you can create visual materials that stand out is in the proper use of VDEPs [9].

Past research has mostly focused on the VDEPs capability of causing specific emotions and how to activate the desired response from the spectator [33,68,69]. A severe limitation of such research has been its inability to boost the classification accuracy of various visual stimuli that are inferred to VDEPs and their impact on human brain activity. Another limitation of previous research has been its inability to demonstrate which cues related to the VDEPs and EEG are most correlated with human brain activity, and the difficulty to reveal the VDEPs that are involved and are conditioning how visual content is perceived.

We sought to address these limitations to understand the relationships between multimedia content itself with users’ physiological responses (EEG) to such content by analysing the VDEPs and their impact on human brain activity.

The results of this study suggest that variations in the light/value (Accuracy 0.90/Loss 0.23), rhythm/movement (Accuracy 0.95/Loss 0,13) and balance (Accuracy 0.81/Loss 0.76) VDEPs in the music video sequences produce a statistically significant effect over the δ, θ, α, β and γ EEG bands, and Colour is the VDEP that has produced the least variation in human brain activity (Accuracy 0.79/Loss 0.50). The CNN model successfully predicts the VDEP of a video fragment solely by the EEG signal of the viewer.

The violin plots confirm that the distributions are practically equal between classes of the same VDEP, and the fact that the *p*-values are significant comes from a large number of samples. We find significant differences in some bands for some VDEPs, but these are minor as the distributions show. However, this could be expected since we are simply analysing power in classical a priori defined bands. However, the CNN results are more remarkable. The fact that we obtain high ranking values with the CNN versus the similarity of the calculated powers in those bands suggests, for future work, that the relationship between VDEPs and EEG is more complex than simple power changes in predetermined frequency bands.

The results show how human brain activity is more susceptible to producing alterations to sudden changes in the visualization of movement, light, or balance in music video sequences than the impact that a colour variation can generate in human brain activity. Reddish tones transmit heat, just as blood is hot and fire burns. The bluish brushstrokes are associated with cold and lightness [70]. These are the colours with which the mind paints, effortlessly and from childhood, water, rain, or a bubble. Different studies support the results obtained on these associations that we can consider universal [68], being directly involved in the design of VDEPs. However, divergences appear when it comes to distinguishing between smooth and rough, male, or female, soft or rigid, aggressive, or calm, for example. Moreover, people disagree more, depending on the results, when judging whether the audio-visual content is interesting, and whether they like it or not.

Brain activity may be slightly altered by changes in the design of different VDEPs such as colour, brightness, or lighting. However, it may be more severely altered when one visualizes sudden changes in movement and course of action. The possible explanation for the results obtained is that the majority of the most basic processes of perception are part of the intrinsic neuronal architecture, the human being more accustomed to this type of modifications in the action that is visualized, and within this context being more susceptible to sudden and abrupt changes in the action that we observe, distorting and altering our brain activity, which has been demonstrated in various studies [69]. Most of the difficulties that arise in studies of this typology are the particularities of the individual. The same person, depending on its perceptual state, will observe the same scene differently, by existing a very wide range of experiences and individual variances [71,72,73,74,75].

Several limitations affect this study. It is important to notice that to the best of our knowledge, there is currently no established ontology for VDEPs. Although the identification of four VDEPs has been sufficient to find and determine the physiological relationships (EEG) and the VDEPs, it would be very interesting if we had a system to organize this knowledge and allow us to construct the architecture of neural networks focused and customized to specific VDEPs, which would permit us to improve accuracy and reduce loss, instead of working with generic networks.

Therefore, we consider that further research in this area should pursue the development of ontologies that let us structure the knowledge related to design, recognition, labelling, filtering, and classification of VDEPs for the improvement and optimization for their use in information systems, expert systems, and decision support systems. Moreover, this study should be extended to research about other VDEPs combinations on brain activity. Future research should also carry out investigations using multiple VDEPs in the EEG of each individual to study the relationships and interactions between different VDEPs. Furthermore, we recommend analysing the VDEPs separately in each EEG channel to understand how the synergies between various VDEPs affects visual perception and visual attention. Finally, future investigations could focus on exploring VDEPs through other models such as generative ones to visualize the topographical distribution of the EEG channels and each VDEP.

## 5. Conclusions

This study found evidence supporting that there is a physiological link between VDEPs and human brain activity. We found that the VDEPs expressed in music video sequences are related to statistically significant differences in the average power of classical EEG bands of the viewers. Furthermore, a CNN classifier tasked to identify the VDEP class from the EEG signals achieved accuracies of 90.44%, 79.67%, 81.25%, and 95.29% of accuracy for the Light, Colour, Balance, and Movement VDEPs, respectively. The results suggest that the relationship between the VDEPs and brain activity is more complex than simple changes in the EEG band power.

## Figures and Tables

**Figure 1 sensors-21-04695-f001:**
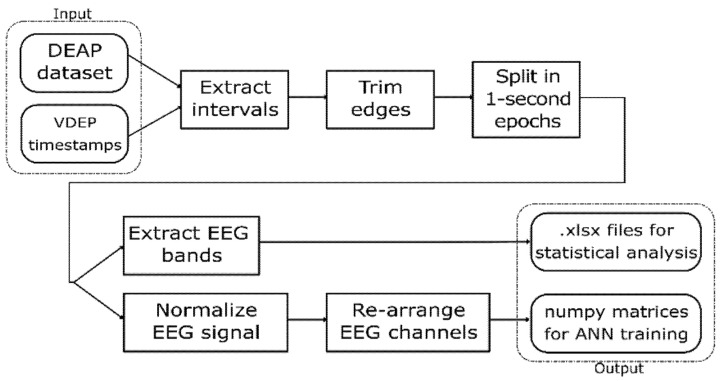
VDEP extraction-transformation process (ETP) from the DEAP dataset.

**Figure 2 sensors-21-04695-f002:**
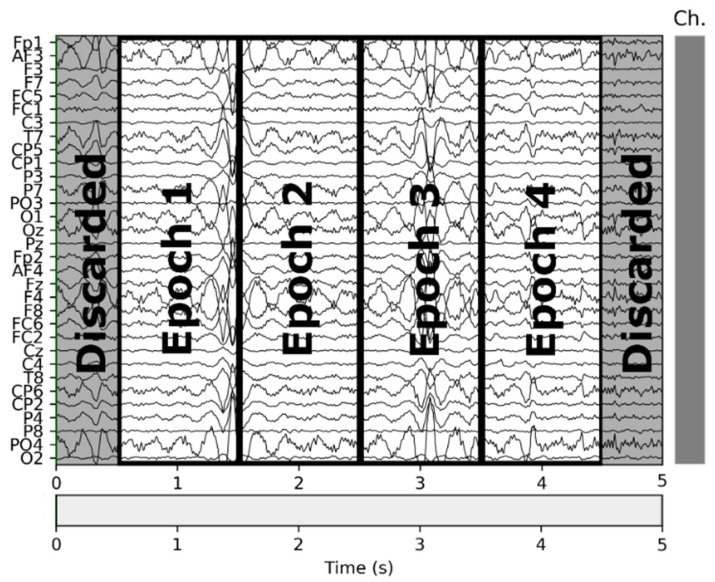
Epochs obtained from a 5-s segment.

**Figure 3 sensors-21-04695-f003:**
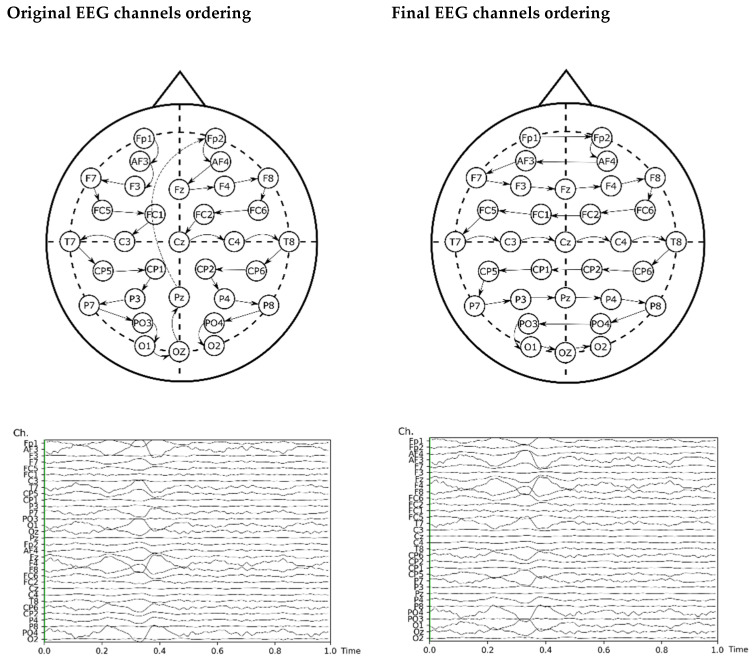
Re-arrangement of the EEG channels for the ANN analysis.

**Figure 4 sensors-21-04695-f004:**
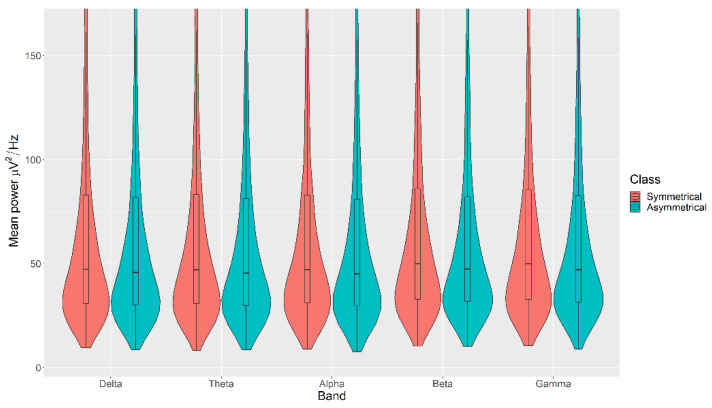
Violin plot of the average power in each of the bands for Symmetrical and Asymmetrical categories for the Balance VDEP, ranging from low (δ delta) to high (γ gamma) frequencies.

**Figure 5 sensors-21-04695-f005:**
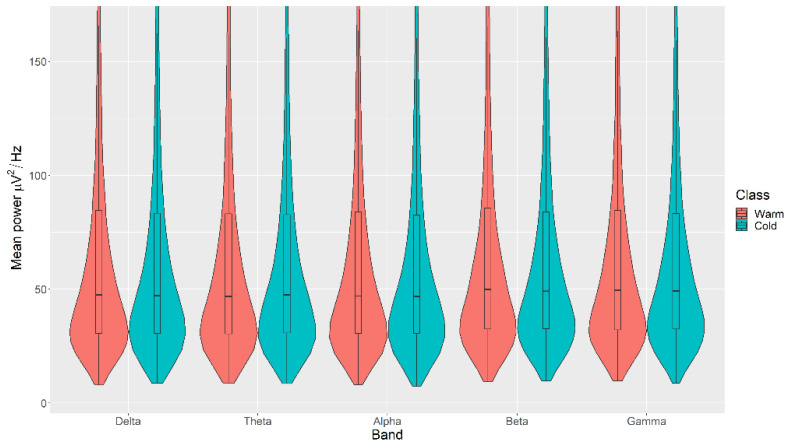
Violin plot of the average power in each of the bands for Warm and Cold categories for the Colour VDEP, ranging from low (δ delta) to high (γ gamma) frequencies.

**Figure 6 sensors-21-04695-f006:**
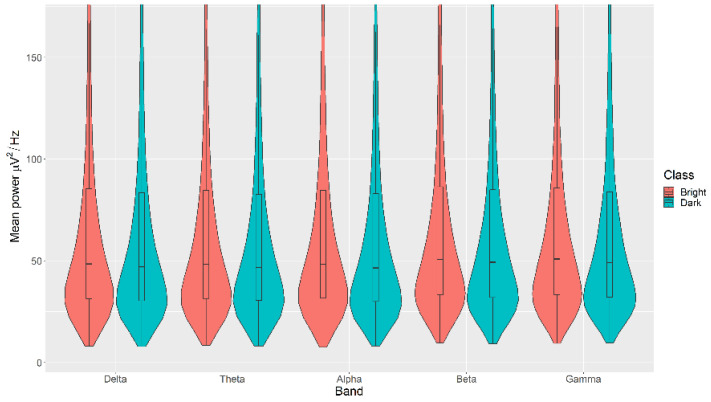
Violin plot of the average power in each of the bands for Bright and Dark categories for the Light VDEP, ranging from low (δ delta) to high (γ gamma) frequencies.

**Figure 7 sensors-21-04695-f007:**
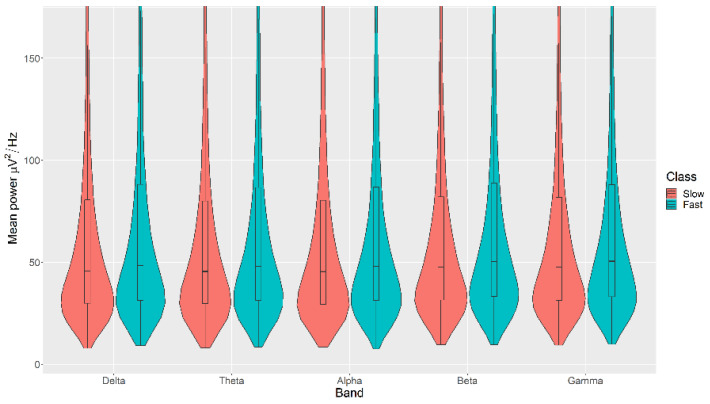
Violin plot of the average power in each of the bands for Slow and Fast categories for the Movement VDEP, ranging from low (δ delta) to high (γ gamma) frequencies.

**Table 1 sensors-21-04695-t001:** The number of video seconds for each VDEP label.

VDEP	Timestamps Class 1	Timestamps Class 2	Unclear Timestamps
Colour	875	588	97
Balance	1276	264	20
Movement	645	535	380
Light	795	654	111

**Table 2 sensors-21-04695-t002:** Performance metrics of the CNN classification model for each of the target VDEPs.

VDEP Targets	AUC	Accuracy	PR-AUC
Light	0.8873	0.8883	0.8484
Colour	0.75560	0.7447	0.6940
Balance	0.7584	0.7477	0.7241
Movement	0.9698	0.9675	0.9569

## Data Availability

Not applicable.

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
