# Peer review of "Impact of Visual Design Elements and Principles in Human Electroencephalogram Brain Activity Assessed with Spectral Methods and Convolutional Neural Networks"

_sensors, 2021, doi:10.3390/s21144695_

Round 1

Reviewer 1 Report

Firstly, I take this opportunity to congratulate the authors on their successful submission of their paper for publication.

The study is well described and executed. However, there are few points that needs to be addressed.

  • Figures: The figure quality and text need to be improved. The units of the label axis (for example, Fig. 5) need to added to the figures.
  • Literature review might need to be improved in terms of papers related to Brain computer interfaces using Convolutional neural networks. The brain computer interface studies related to Steady State Evoked Potentials closely resemble with the Fast/Slow movement in the current analysis. The following are few papers you may refer to:

Vialatte, F.B., Maurice, M., Dauwels, J. and Cichocki, A., 2010. Steady-state visually evoked potentials: focus on essential paradigms and future perspectives. Progress in neurobiology90(4), pp.418-438.

Thomas, J., Maszczyk, T., Sinha, N., Kluge, T. and Dauwels, J., 2017, October. Deep learning-based classification for brain-computer interfaces. In 2017 IEEE International Conference on Systems, Man, and Cybernetics (SMC) (pp. 234-239). IEEE.

Kwak, N.S., Müller, K.R. and Lee, S.W., 2017. A convolutional neural network for steady state visual evoked potential classification under ambulatory environment. PloS one12(2), p.e0172578.

Stober, S., Cameron, D.J. and Grahn, J.A., 2014. Using Convolutional Neural Networks to Recognize Rhythm Stimuli from Electroencephalography Recordings. In Advances in neural information processing systems (pp. 1449-1457).

  • CNN evaluation: While evaluating a machine learning or deep learning model, cross-validation should be performed in order to validate the efficacy of the model. The data should be divided into multiple folds, train/test evaluate the model multiple times, and average the results. Also, the data from a single patient should not be separated across multiple folds.
  • Performance measures: Accuracy is a skewed metric for evaluating the performance of a model. Since this is class-imbalanced problem, you need to explore measures such as Area Under curve, Area under precision recall curve, etc., to represent and evaluate the results. The studies I had suggested previously details the different performance metrics.
  • Baseline performance: Since an objective of this study is to explore the efficiency of a CNN for classification, in order to verify whether a CNN is required, the baseline performance is to be computed using traditional classifiers such as SVM, decision trees, etc.
  • Confusion regarding the performance of colour : The CNN performed with an accuracy of ~80% even though the features were not statistically significant. Therefore, this result needs to be verified.

Author Response

Firstly, I take this opportunity to congratulate the authors on their successful submission of their paper for publication.

Dear reviewer,

We are very pleased that you liked the manuscript and very grateful that you took the time to review it.

The study is well described and executed. However, there are few points that needs to be addressed.

  • Figures: The figure quality and text need to be improved. The units of the label axis (for example, Fig. 5) need to added to the figures.

Thank you very much for your feedback. The units of the label axis (Figure 4,5,6 and 7) were added to the figures. Although the images included in the manuscript are in High-res, during the submission process, due to some compression adjustment of the MDPI templates, the images appear compressed in Low-res, to avoid this, the figures have been included separately in the resubmission in TIFF format (300dpi). 

  • Literature review might need to be improved in terms of papers related to Brain computer interfaces using Convolutional neural networks. The brain computer interface studies related to Steady State Evoked Potentials closely resemble with the Fast/Slow movement in the current analysis. The following are few papers you may refer to:

Vialatte, F.B., Maurice, M., Dauwels, J. and Cichocki, A., 2010. Steady-state visually evoked potentials: focus on essential paradigms and future perspectives. Progress in neurobiology90(4), pp.418-438.

Thomas, J., Maszczyk, T., Sinha, N., Kluge, T. and Dauwels, J., 2017, October. Deep learning-based classification for brain-computer interfaces. In 2017 IEEE International Conference on Systems, Man, and Cybernetics (SMC) (pp. 234-239). IEEE.

Kwak, N.S., Müller, K.R. and Lee, S.W., 2017. A convolutional neural network for steady state visual evoked potential classification under ambulatory environment. PloS one12(2), p.e0172578.

Stober, S., Cameron, D.J. and Grahn, J.A., 2014. Using Convolutional Neural Networks to Recognize Rhythm Stimuli from Electroencephalography Recordings. In Advances in neural information processing systems (pp. 1449-1457).

Thank you very much for your valuable inputs. We have expanded the information about the SSVEPS as well as its relationship with the BCI and CNNs and included the suggested references.

  • CNN evaluation: While evaluating a machine learning or deep learning model, cross-validation should be performed in order to validate the efficacy of the model. The data should be divided into multiple folds, train/test evaluate the model multiple times, and average the results. Also, the data from a single patient should not be separated across multiple folds.

Thank you for pointing this out. We performed 10-fold cross-validation that yielded similar results. The related changes are now indicated in sections 2.3 and 3.5. However, we could not isolate the data from a single patient to not being separated across folds because there is a very small number of patients (32) related to the number of folds. In this regard, we will take this into account for further research that analyses each patient in a more in-depth approach.

  • Performance measures: Accuracy is a skewed metric for evaluating the performance of a model. Since this is class-imbalanced problem, you need to explore measures such as Area Under curve, Area under precision recall curve, etc., to represent and evaluate the results. The studies I had suggested previously details the different performance metrics.

Thank you for this suggestion. In the revised manuscript we provide the AUC and the PR-AUC metrics related to the 10-fold cross-validation experiment.

  • Baseline performance: Since an objective of this study is to explore the efficiency of a CNN for classification, in order to verify whether a CNN is required, the baseline performance is to be computed using traditional classifiers such as SVM, decision trees, etc.

The aim of this research was not to explicitly say that a CNN model is needed. In fact, in section 2.3 we stated that we choose a generic CNN, that comes without any optimization or fine tunning for this specific classification problem, to prove that it is possible to identify the VDEPs of visual information exposed to the viewer by analysing solely the EEG signal. Further research can and must be done to create more sophisticated models for this problem. In this regard, it is highly possible that a traditional classifier such as SVM can achieve similar performance to the simple CNN model that we used, but adding it to this paper would be beyond its scope.

  • Confusion regarding the performance of colour : The CNN performed with an accuracy of ~80% even though the features were not statistically significant. Therefore, this result needs to be verified.

Thank you for pointing this out. We double-checked the results of the cross-validation experiment and confirmed that the CNN is able to classify most of the samples of Colour. The statistical analysis indicated that there was no difference between the mean power of the bands for the Colour VDEP. The fact that the CNN is able to find these differences hints that the relationship between VDEPs and EEG is more complex than simple power changes in predetermined frequency bands.

Reviewer 2 Report

Thank you for recommending me as reviewer. In this study, the authors were explore the relationships between brain activity and the colour (cold/warm), light (dark/bright), movement (fast/slow) and balance (symmetrical/asymmetrical) VDEPs. The author was used the public DEAP Dataset with electroencephalogram signals of 32 participants recorded while watching music videos. If authors complete minor revisions, the quality of the study will be further improved.

  1. The introduction section is well written. The Introduction section includes overview and background.

2. The methods section is written clearly and specifically.

3. line 326-327:  Is there any particular reason for using Mann-Whitney U in this paper? The author should add a description of the statistical test to the research method section.

4. 374: Discussion: Why is the prediction accuracy of Color VDEP (0.79) so low in this paper? If precision and recall were analyzed separately in addition to accuracy, wouldn't the results be different?

5. In Table A1, etc., it is appropriate to express the significance probability .000 as <0.001.

Author Response

Thank you for recommending me as reviewer. In this study, the authors were explore the relationships between brain activity and the colour (cold/warm), light (dark/bright), movement (fast/slow) and balance (symmetrical/asymmetrical) VDEPs. The author was used the public DEAP Dataset with electroencephalogram signals of 32 participants recorded while watching music videos. If authors complete minor revisions, the quality of the study will be further improved.

 Dear reviewer,

We are very pleased that you liked the manuscript and very grateful that you took the time to review it.

  1. The introduction section is well written. The Introduction section includes overview and background.
  2. The methods section is written clearly and specifically.
  3. Line 326-327:  Is there any particular reason for using Mann-Whitney U in this paper? The author should add a description of the statistical test to the research method section

Thank you for pointing this out. In this case, the Mann-Whitney U test was chosen to test the null hypothesis that there is a probability of 0.5 that a randomly drawn observation for one group is larger than a randomly drawn observation from the other. We could have also used the t-test with similar results, but decided to stick with the Mann-Whitney U test as it followed more closely our objective. This is now indicated in the manuscript. The term “mean band measurements” might have caused confusion, but our goal was not to compare the means of two distributions, but to compare two distributions of mean power measurements.

  1. 374: Discussion: Why is the prediction accuracy of Color VDEP (0.79) so low in this paper? If precision and recall were analyzed separately in addition to accuracy, wouldn't the results be different?

We performed a 10-fold cross-validation experiment and computed the AUC and Area under the precision-recall curve (PR-AUC) in addition to the accuracy. The Colour VDEP classifier achieved the following mean performance metrics: AUC = 0.7584, Accuracy = 0.7447, PR-AUC = 0.6940. We did not expect such a low performance on the prediction of Colour VDEP, but it is also important to notice that we used a simple generic-purpose CNN model. These results suggest that the alteration in the brain activity produced by the Colour VDEP is small, or not limited to variations in the mean power of EEG bands.

  1. In Table A1, etc., it is appropriate to express the significance probability .000 as <0.001.

We reviewed and changed the significance probability in those cases.
